# Establishment and Characterization of SV40 T-Antigen Immortalized Porcine Muscle Satellite Cell

**DOI:** 10.3390/cells13080703

**Published:** 2024-04-18

**Authors:** Mengru Ni, Jingqing He, Tao Li, Gan Zhao, Zhengyu Ji, Fada Ren, Jianxin Leng, Mengyan Wu, Ruihua Huang, Pinghua Li, Liming Hou

**Affiliations:** 1College of Animal Science and Technology, Nanjing Agricultural University, Nanjing 210095, China; 2021105009@njau.edu.cn (M.N.); 2021105014@njau.edu.cn (J.H.); 2021105011@njau.edu.cn (T.L.); 2022105028@njau.edu.cn (G.Z.); jzy604672583@gmail.com (Z.J.); 35121204@njau.edu.cn (F.R.); 35121112@njau.edu.cn (J.L.); 35121207@njau.edu.cn (M.W.); rhhuang@njau.edu.cn (R.H.); lipinghua718@njau.edu.cn (P.L.); 2Institute of Swine Science, Nanjing Agricultural University, Nanjing 210095, China; 3Key Laboratory of Pig Genetic Resources Evaluation and Utilization (Nanjing) of Ministry of Agriculture and Rural Affairs, Nanjing Agricultural University, Nanjing 210095, China; 4Huai’an Academy, Nanjing Agricultural University, Huai’an 223001, China

**Keywords:** SV40 T-antigen, muscle satellite cells, pig, immortalization, stemness maintenance

## Abstract

Muscle satellite cells (MuSCs) are crucial for muscle development and regeneration. The primary pig MuSCs (pMuSCs) is an ideal in vitro cell model for studying the pig’s muscle development and differentiation. However, the long-term in vitro culture of pMuSCs results in the gradual loss of their stemness, thereby limiting their application. To address this conundrum and maintain the normal function of pMuSCs during in vitro passaging, we generated an immortalized pMuSCs (SV40 T-pMuSCs) by stably expressing SV40 T-antigen (SV40 T) using a lentiviral-based vector system. The SV40 T-pMuSCs can be stably sub-cultured for over 40 generations in vitro. An evaluation of SV40 T-pMuSCs was conducted through immunofluorescence staining, quantitative real-time PCR, EdU assay, and SA-β-gal activity. Their proliferation capacity was similar to that of primary pMuSCs at passage 1, and while their differentiation potential was slightly decreased. SiRNA-mediated interference of SV40 T-antigen expression restored the differentiation capability of SV40 T-pMuSCs. Taken together, our results provide a valuable tool for studying pig skeletal muscle development and differentiation.

## 1. Introduction

Muscle satellite cells (MuSCs) are a major cell type involved in skeletal muscle regeneration and are essential for the growth, maintenance, and repair of skeletal muscle postnatally [1,2,3]. These cells exist in a quiescent state between the sarcolemma and the basal lamina of muscle fibers, exhibiting heterogeneity [2]. In mature resting muscle, MuSCs mainly remain in a quiescent state. However, upon muscle damage, these cells are activated; and undergo self-renewal, proliferation, differentiation, and fusion. Ultimately, they will form new muscle fibers and repair damaged muscle tissue [4,5,6].

Research has shown that paired box protein 7 (*PAX7*) serves as a marker gene and important regulatory factor for satellite cells [7]. The differentiation ability of MuSCs is controlled by myogenic regulatory factors (MRFs), such as myogenic factor 5 (MYF5), myogenic differentiation factor (MYOD), myogenic regulatory factor 4 (MRF4), and myogenin (MYOG) [8]. MYOG is responsible for driving terminal differentiation and activating the expression of the target genes necessary for this process [9]. Upon activation, the expression of *MYF5* and *MYOD* rapidly increased [10,11]. Subsequently, myogenic cells enter the final differentiation stage, expressing *MYOG*, *MRF4*, and other genes specific to mature muscle fibers, such as myosin heavy chain (*MyHC*). Therefore, the expression of *PAX7*, *MYOD*, and *MYOG* can be used to distinguish different stages of MuSCs, including *PAX7*+*MYOD-* (quiescent/self-renewal), *PAX7*+*MYOD*+ (proliferation), and *PAX7-MYOD*+*MYOG*+ (differentiation) [12,13].

Pigs have many advantages in modeling human diseases due to their similar anatomic and physiological features to human beings. Pig MuSCs (pMuSCs) are crucial materials for studying skeletal muscle development and regeneration and also can be used to produce cultured meat [14]. The characterization of MuSCs in pigs will be a valuable addition to our understanding of the porcine model system. Like MuSCs from other species, pig MuSCs also lose their proliferation and differentiation abilities during long-term in vitro culture [15]. Furthermore, the isolation of primary cells is associated with drawbacks such as time consumption, labor-intensive procedures, and significant inter-individual variations. Researches on pig muscle often limited by using primary MuSCs. Therefore, it is necessary to establish porcine muscle satellite cells capable of long-term stable passaging in vitro.

One common method to inhibit cell apoptosis is using of simian virus 40 large T-antigen (SV40 T-antigen) expression constructs [16]. In in vitro cell cultures, SV40 T-antigen can facilitate the transformation of oncogenes. The T-antigen interacts with tumor-suppressor proteins Rb (retinoblastoma) and p53, forming complexes with the Rb-binding domain and p53-binding domain. This interaction inhibits the tumor-suppressing effects of Rb and p53, thereby relieving their inhibition on the cell cycle. This leads to accelerated cell division and unlimited growth [17]. In most cases, the T-antigen induces the inactivation of Rb protein, leading to the activation of E2F-dependent transcription and subsequent entry into the S-phase. Additionally, the antigen inhibits the activity of p53, thereby preventing subsequent cell cycle arrest and apoptosis [18].

In recent years, scientists have developed various immortalized cell lines. The currently known immortalized pig cell lines mainly comprise the following: PK15 (Porcine Kidney 15), a commonly used pig kidney cell line that is widely applied in virus research, vaccine production, and biopharmaceutical fields [19]. LLC-PK1 (Lilly Laboratory Porcine Kidney 1), derived from pig kidney tissue, is frequently utilized to investigate drug metabolism, toxicity, and cellular biology [20]. IPEC-J2 (Intestinal Porcine Epithelial Cell-J2), derived from pig small intestinal epithelial cells, is commonly employed to study pig intestinal physiology, diseases, and nutrition [21]. In addition to these cell lines, there are also other cell lines utilized in pig including pig embryonic cell lines, fibroblast cell lines, and lymphoblast cell lines [22,23]. These cell lines play a significant role in studying various aspects of pig health, reproduction, immunity, and nutrition. However, no relevant reports on pig satellite cell lines have been found to date.

In this study, primary MuSCs was isolated from a 1-day-old Western commercial pig’s longissimus dorsi muscle using the differential adhesion method. Subsequently, a pig MuSCs line was established by transducing the SV40 T-antigen. The established SV40 T-antigen pig muscle satellite cells (SV40 T-pMuSCs) could provide a valuable source of cells for studying pig skeletal muscle development and differentiation. 

## 2. Materials and Methods

### 2.1. Pig Primary MuSCs Isolation and Differentiation

Pig muscle primary satellite cells (pMuSCs) were isolated from the longissimus dorsi muscle of one 1-day-old Western commercial pig using the differential adhesion method. Briefly, the longissimus dorsi muscle on both sides was minced and incubated with 0.2% type I collagenase (Sigma, Darmstadt, Germany, Cat# SCR103) (twice the volume) at 37 °C and 100 rpm in a water bath for 2–2.5 h until the solution became a yellow viscous substance. After centrifugation to remove the collagenase I, the cell pellet was resuspended in 0.25% trypsin–EDTA (Gibco, Glen Island, New York, NY, USA, Cat# 25200072) (three times the volume) and digested at 37 °C and 100 rpm for 30 min. An equal volume of growth medium (GM) was added to stop the digestion, and the cells were successively filtered through 100 μm and 70 μm cell strainers. The GM medium consisted of DMEM (Gibco, Carlsbad, CA, USA, Cat# 6123118) supplemented with 10% Fetal Bovine Serum (FBS) (Dcell, Darmstadt, Germany, Cat# 23055525) and 1% penicillin–streptomycin (Gibco, Carlsbad, CA, USA, Cat# 15140122). The supernatant was removed by centrifugation at 2000 rpm for 12 min. The cells were resuspended in PBS, centrifuged at 1500 rpm for 10 min to remove the PBS, and this step was repeated twice. Finally, the cells were resuspended in complete GM medium and incubated for 2 h on three of 25 cm^2^ cell culture flasks. The complete GM medium consisted of F12 (Gibco, Grand Island, NE, USA, Cat# 11330032) supplemented with 20% FBS (Dcell, Darmstadt, Germany, Cat# 23055525), 1% penicillin–streptomycin (Gibco, Carlsbad, CA, USA, Cat# 15140122), 1% GlutaMAX (Gibco Grande Island, New York, NY, USA Cat# 35050079), 1% sodium pyruvate (Gibco, Grande Island, NY, USA, Cat# 11360070), 1% non-essential amino acid (Gibco, Grande Island, NY, USA, Cat# 11140050), and 2 ng/mL bFGF (Peprotech, Rocky Hill, NJ, USA, Cat# 450-33). The supernatant was collected as satellite cells and cultured in 37 °C, 25 cm^2^ cell culture flasks, with the medium being replaced every 48 h. The pMuSCs were differentiated for 5 days, the differentiation medium consisted of DMEM supplemented with 0.4% Ultroser G (SARTORIUS, Berlin, Germany, Cat# 15950-017) and 1% penicillin–streptomycin (Gibco, Carlsbad, CA, USA, Cat# 15140122). For the statistical analysis, the differentiation index is defined as the percentage of MyHC+ nuclei compared to the total number of nuclei, while the myotube fusion index is defined as the ratio of the total number of nuclei within myotubes (consisting of three or more nuclei) to the total number of MyHC+ nuclei.

### 2.2. Immunofluorescence Staining

The cells with designated generation were plated in a 24-well plate, and the experiment was initiated once the confluence reached 50%. Firstly, the cell culture medium was removed, and the cells were washed with PBS and fixed for 30 min using 4% paraformaldehyde. Then, the cells were permeabilized with 0.3% Triton X-100 for 10 min and washed three times with PBS. Quick Blocking solution (Beyotime, Shanghai, China, Cat# P0260) was applied and incubated for 1 h. The primary antibody was added and incubated overnight at 4 °C, followed by PBS wash for three times. The secondary antibody was added and incubated at room temperature in dark for 1 h. After removing the secondary antibody, the cells were washed with PBS three times, and the cell nuclei were stained using anti-fade mounting medium with DAPI (Bioshrap, Shanghai, China, Cat# BL739A). The experiments were repeated three times. Images were captured using a fluorescence Nexcope NIB610-FL microscope, and representative images were shown. Primary antibodies used in this study include anti-MyHC (MF-20, 1:100) (Developmental Studies Hybridodma Bank, Ithaca, New York, NY, USA, Cat# AB_2147781), anti-PAX7 (Developmental Studies Hybridodma Bank, Ithaca, New York, NY, USA, Cat# AB_528428), and anti-SV40 T-antigen (Abcam, Cambridge, UK, Cat# ab234426). Secondary antibodies used in this study include Cy3-labeled goat anti-mouse (1:1000) (Beyotime, Shanghai, China, Cat# A5021), Alexa Fluor 488-labeledgoat anti-mouse (Beyotime, Shanghai, China, Cat# A0428), and Alexa Fluor 488-labeled goat anti-rabbit (Beyotime, Shanghai, China, Cat# A0423).

### 2.3. Real-Time Quantitative PCR (qPCR)

The total RNAs were extracted using TRIzol (Ambion, Carlsbad, CA, USA, Cat# 380512). cDNA was synthesized using an Evo M-MLV Mix Tracking Kit with gDNA Clean (Accurate, Shanghai, China, Cat# AG11734), following the manufacturer’s instructions. qPCR was performed using Hieff UNICON Universal Blue qPCR SYBR Green Master Mix (Yeasen, Shanghai, China, Cat# 11141ES) on a CFX96 Optics Module. The PCR program is as follows: pre-denaturation—95 °C, 2 min; denaturation—95 °C, 10 s; and annealing—60 °C, 30 s. Cycle number is 40. The primers used in these study were as follows: *PAX7*-F, 5′-GTGCCCTCAGTGAGTTCGATT-3′, *PAX7*-R, 5′-TCCAGACGGTTCCCTTTGTC-3′; *MYOD*-F, 5′- GCTCCGCGACGTAGATTTGA-3′, *MYOD*-R, 5′-GGAGTCGAAACACGGGTCAT-3′, *SV40 T-antigen*-F: GTGCCTAAAACACTGCAGGC, *SV40 T-antigen*-R: CAGCCACAGGTCTGTACCAA. *GAPDH*-F, 5′-TGAGATCAGGGAGCCATCA-3′, *GAPDH*-R, 5′ATGGTCAGGGGTCCGAT-GTA-3′. The relative expression levels of the target gene were calculated using the 2^−ΔΔCt^ method. The experimental results are presented as the mean ± standard error (SEM). Statistical analysis and tests for significant differences among groups were performed using IBM SPSS Statistics 26 software.

### 2.4. Transduction of SV40 T-Antigen to pMuSCs

Lentiviral particles containing the SV40 T-antigen were obtained using human 293T cells. The 293T cells reached 70% density in a six-well plate were transfected with a lipofectamine 2000 DNA reagent. LV-EF1a-SV40-Large-T-Antigen-P2A-Hygro-Barcode (Addgene, Watertown, MA, USA, Cat# 170255) (1 μg), psPAX2 (Addgene, Watertown, MA, USA, Cat# 12260) (750 ng), and pMD2.G (Addgene, Watertown, MA, USA, Cat# 12259) (250 ng) were co-transfected into the cells. The supernatants from the cells were collected at both 24 h and 48 h, and the supernatants can be used as lentiviral particle solutions after filtration through a 0.45 μm filter, and then stored at −80 °C for later use. pMuSCs were grown in a six-well plate to reach 20% density, then added 2 mL of complete GM medium with 500 μL SV40 T lentiviral particles and 8 μg/mL polybrene. After 24 h of infection, the medium was changed, and the cells were allowed to rest for one day. Subsequently, the cells were cultured in complete GM medium containing 200 μg/mL hygromycin B for 7 days to select the positive cells expressing SV40 T-antigen (SV40T-pMuSCs).

### 2.5. EdU Assay

EdU assay was conducted using pMuSCs and SV40 T-pMuSCs at different passages. The 5-ethynyl-2′-deoxyuridine (EdU) assay was performed according to the manufacturer’s protocol (Beyotime, Shanghai, China, Cat# C0078S). The cells were cultured in 12-well plates. The experiment was initiated when the cell confluence reached 50%. A 2× working solution of EdU was prepared in complete GM from the 10 mM stock solution. Then, 500 μL 2× EdU working solution was added to each well, resulting in 1× EdU (10 μM). After 2 h of incubation, cells were fixed by 4% paraformaldehyde (PFA) for 15 min. Subsequently, cells were permeabilized in 0.3% Triton X-100 in PBS for 15 min and stained with Click reaction solution for 30 min. After washing, the nuclei were stained with DAPI for 10 min. Cells were imaged with a Nexcope NIB610-FL microscope. Around 1000 randomly elected cells were analyzed.

### 2.6. SA-β-Gal Activity

The SA-β-gal activity was assessed in cells from different passages, including pMuSCs (P1, P2, P4, and P10) and SV40 T-pMuSCs (P1, P10, P30, and P40). A senescence β-Galactosidase Staining Kit (Beyotime, Shanghai, China, Cat# C0602) was used according to the manufacturer’s instruction. The cells were cultured in 12-well plates and fixed with stationary liquid for 15 min. Subsequently, cells were washed once with PBS. A volume of 500 µL β-galactosidase staining solution was added, and the cells were incubated overnight at 37 °C in a CO_2_-free chamber. The resulting images were directly captured using a Nexcope NIB610-FL microscope.

### 2.7. RNA-seq Analysis

RNA samples were collected from P1, P2, and P4 generations of pMuSCs using TRIzol. RNA-seq libraries were generated by Nanopore Co., Ltd. (Wuhan, China). The libraries were sequenced using the Illumina NovaSeq 6000. For the data analysis, fastp (v0.23.2) was used for quality filtering of the sequencing reads, and reads with unknown sequences >10% and quality scores <20 were removed. STAR (v2.7.9) was used for a fast and accurate sequence aligned to the Suscrofa11.1 reference genome. Finally, a transcriptome gene expression count file was converted using feature Counts (v2.0.3) to obtain the gene expression profile in each sample. Differentially expressed genes were identified by DESeq2 (v.1.20). Genes with a corrected *p*-value < 0.05 and log2 fold changes > 1 were assigned as significantly differentially expressed.

### 2.8. SiRNA Transfection

The siRNAs were designed and synthesized by Gene Pharma. Cells were plated into two 12-well plates for the siRNA interference assay, with three replicates per group. The experiment was initiated when the cells reached a confluence of 50%. SV40 T-pMuSCs were transfected with either 50 nM siRNA (si-SV40 T) or control (si-NC) using Lipofectamine RNAiMAX (Invitrogen, Carlsbad, CA, USA, Cat# 13778150). Cells were collected 48 h post-transfection, and RNAs were extracted using the aforementioned method and reverse transcribed for qPCR. After 48 h of transfection, the differentiation medium was replaced, and cells were induced to differentiate for 3 days. The sequences of the siRNAs used were as follows: si-NC sense (5′-3′): UUCUCCGAACGUGUCACGUTT, si-NC antisense (5′-3′): ACGUGACACGUUCGGAGAATT si-SV40 T-antigen-1 sense (5′-3′): GGAGUUUCAUCCUGAUAAATT, si-SV40 T-antigen-1 antisense (5′-3′): UUUAUCAGGAUGAAACUCCTT; si-SV40 T-antigen-2 sense (5′-3′): GUGGUGGAAUGCCUUUAAUTT, si-SV40 T-antigen-2 antisense (5′-3′): AUUAAAGGCAUUCCACCACTT.

### 2.9. Statistical Analysis

All data were expressed as the mean ± standard error of mean (SEM). Statistical analysis was performed using GraphPad Prism 8. For comparisons of two treatment groups, a *t*-test was used. For more than two groups, one-way ANOVA with Tukey’s multiple comparison tests was used. Statistical significance was set at * *p* < 0.05 or ** *p* < 0.01.

## 3. Results

### 3.1. Evaluation of Function of pMuSCs during Long-Term In Vitro Culture

In order to evaluate the stemness of pMuSCs over in vitro culturing, we assessed the expression of the satellite cell marker gene *PAX7* across different passages of pMuSCs. The immunofluorescence staining revealed that, at P1, approximately 65.7 ± 6.0% of pMuSCs were PAX7+, which decreased to approximately 20.86 ± 1.6% at P10 (Figure 1A). qPCR results indicated a decline in the expression levels of *PAX7* and *MYOD* with increased cell passage number (Figure 1B,C).

To further explore the effect of in vitro culture on the differentiation ability of pMuSCs, the cells were induced to differentiate at P1, P2, P4 and P10 by switching into the differentiation medium when they reached 90% confluency. The differentiation ability was indicated by myosin heavy chain (MyHC) staining. MyHC+ cells decreased from 55.5 ± 9.3% at P1 to 2.6 ± 1.1% at P10 (Figure 1C). Altogether, pMuSCs lost their stemness and differentiation ability during prolonged in vitro culture.

Furthermore, the EdU incorporation results showed that cells at late passage had a lower proliferation rate (Figure 1D). We also assessed cell senescence for pMuSCs cells at different passages by senescence-associated β-galactosidase (SA-β-gal) staining (Figure 1E). The percentage of senescent cells gradually increased to about 33.3 ± 6.1% in P10. In conclusion, these results suggested that, with the increased passage, the proliferation rate and differentiation potential of pMuSCs were reduced, and the cellular senescence rate increased.

### 3.2. Evaluate the Transcriptional Profile of Different Passage of pMuSCs

To understand the reason why pMuSCs lost normal proliferation and differentiation function during long-term in vitro culture, we collected mRNAs from P1, P2, and P4 pMuSCs cells for RNA-Seq analysis. Differentially expressed genes (DEGs) were identified based on the criteria of Padj < 0.001 and log2 (fold change) ≥ 1. When comparing P2 with P1 generation of pMuSCs, a total of 274 DEGs were identified, with only 94 downregulated genes and 180 upregulated genes (Figure 2A, Appendix A), suggesting that the difference between these two generations of cells was not so obvious, which is consistent with our previous findings from the PAX7 and MyHC immunofluorescence assays (Figure 1A,C). When comparing P4 with P1 generation of pMuSCs, a total of 4341 DEGs were identified, including 2166 upregulated genes and 2175 downregulated genes (Figure 2B, Appendix A). When comparing P4 with P2 generation of pMuSCs, a total of 3730 DEGs were identified, comprising 1815 upregulated genes and 1915 downregulated genes (Figure 2C, Appendix A). These results indicated a substantial alteration in the gene expression pattern of pMuSCs during long-term in vitro culture.

Based on the KEGG (Kyoto Encyclopedia of Genes and Genomes) pathway enrichment analysis comparing the P2 and P1 generations of pMuSCs, we discovered that the upregulated DEGs were predominantly enriched in pathways such as motor proteins, Cytokine-cytokine receptor interaction, and mTOR signaling pathway (Figure 2D). Conversely, the downregulated DEGs were primarily enriched in pathways including cell adhesion molecules, Rap1 signaling pathway, PI3K-Akt signaling pathway, and cAMP signaling pathway (Figure 2E). Based on the comparison between the P4 and P1 generations of pMuSCs, we found that the upregulated DEGs were primarily enriched in pathways such as the MAPK signaling pathway, NF-kappa B signaling pathway, and FoxO signaling pathway (Figure 2F). Conversely the Conversely, the downregulated DEGs were primarily enriched in pathways related to DNA replication, base excision repair, and mismatch repair (Figure 2G). When comparing the P4 with P2 generation of pMuSCs, the upregulated DEGs were mainly enriched in pathways such as PI3K-Akt signaling pathway, NOD-like receptor signaling pathway, and Rap1 signaling pathway (Figure 2H). And the downregulated DEGs were primarily enriched in pathways related to DNA replication, base excision repair, mismatch repair, and cell cycle (Figure 2I). The enrichment pathways for downregulated genes between the P4 and P1 generations of pMuSCs are similar to those found to be enriched in the comparison between P4 and P2, with all of them showing enrichment in signaling pathways such as DNA replication, base excision repair, and mismatch repair. These results suggest that, as pMuSCs underwent long-term in vitro culture, their abilities in DNA replication, base excision repair, and nucleotide excision repair gradually declined. This decrease may ultimately result in cellular senescence phenotypes characterized by reduced proliferation and differentiation capacities.

### 3.3. Establishment and Characterization of SV40 T-pMuSCs

A common method to establish an immortalized cell line is to transfect the primary cells with SV40 T-antigen gene [24]. In this study, we transduced primary pMuSCs with a lentiviral vector expressing SV40 T-antigen. The entire procedure for establishing an in vitro sustainable passaging porcine MuSC line was illustrated in Appendix A. 

To confirm the stable expression of the exogenous SV40 T-antigen gene during the passaging of SV40 T-pMuSCs, SV40 T immunofluorescence staining was performed on the P1, P10, P30, and P40 generations of SV40 T-pMuSCs (Appendix A). There were no significant differences in the positive rate of SV40 T-antigen among different generations of SV40 T-pMuSCs, indicating the stable expression of SV40 T-antigen. PAX7 immunofluorescence staining was performed on the aforementioned SV40 T-pMuSCs with different generations. Interestingly, the PAX7+ rate in the P1 generation of SV40 T-pMuSCs was relatively low (Figure 3A). However, as the cells underwent passaging, the percentage of PAX7+ cells gradually increased and eventually stabilized at around 60%, similar to the PAX7+ rate seen in P1 generation of pMuSCs (Figure 1A). 

In addition, we found that SV40 T-pMuSCs can be cultured in vitro for up to 40 generations, while maintaining the morphology consistent with the P1-generation of pMuSCs (Appendix A). We further performed EdU incorporation assays on different generations of SV40 T-pMuSCs and found no significant differences in the proliferation rates among them. The number of EdU+ cells was similar to that of the P1-generation of pMuSCs (Figure 1D and Figure 3B), indicating that SV40 T-antigen can maintain the proliferation capacity of the cells. Subsequently, we performed β-galactosidase staining on P1, P10, P30, and P40 generations of SV40 T-pMuSCs to assess the cellular senescence. Notably, as the cells underwent passaging, the β-galactosidase positive rate gradually increased, indicating that SV40 T-antigen could not prevent cellular senescence (Figure 3C). 

To further validate the functionality of SV40 T-pMuSCs, we induced the differentiation of P1-, P2-, P4-, P10-, P30-, and P40-generation SV40 T-pMuCs in differentiation medium. The expression of MyHC indicated the ability for myotube formation. The immunofluorescence results showed that the percentage of MyHC+ cells in each generation was around 30% (Figure 3D), which was lower than that in P1-generation pMuSCs (Figure 1C, around 55.5 ± 9.3%. And there were no significant differences in MyHC+ rates among different generations of SV40 T-pMuCs, indicating that SV40 T-antigen can maintain the differentiation ability of pMuSCs. Moreover, we observed that the myotubes formed from SV40 T-pMuSCs were visibly thinner compared to those formed from primary pMuSCs. Furthermore, we calculated the fusion index, and found that for both pMuSCs and SV40 T-pMuSCs after differentiation, the fusion index of SV40 T-pMuSCs was significantly lower than that of pMuSCs (Figure 3E). These results suggested that while the SV40 T pMuSCs could maintain the proliferation ability in vitro, consistently expressed SV40 T-antigen blocked the differentiation ability of pMuSCs.

### 3.4. Interfering SV40 T-Antigen Gene Restored the Differentiation Capacity of SV40 T-pMuSCs

To investigate the impact of SV40 T-antigen on cell differentiation capacity, we transfected P1 and P30 generations of SV40 T-pMuSCs (which represent the low and high passages, respectively) with siRNA targeting SV40 T-antigen. The qPCR showed a significant decrease in the expression level of *SV40 T-antigen* gene after interference (Figure 4A). Subsequently, the cells were induced to differentiate for 5 days in a differentiation medium. We quantified the number of MyHC+ cells and the myotube fusion index. Both low- and high-passage cells showed significantly higher numbers of MyHC+ cells and myotube fusion indices compared to the negative control (NC) group after interfering with SV40 T-antigen (Figure 4B,C). These results demonstrated that interfering with SV40 T-antigen can restore the differentiation capacity of SV40 T-pMuSCs.

## 4. Discussion

Muscle satellite cells were initially discovered by Mauro in 1961 and were believed to play a role in skeletal muscle growth and regeneration [1]. Muscle satellite cells, responsible for muscle regeneration, serve as crucial materials for studying skeletal muscle development and regeneration. It has been reported that the regenerative capacities of mouse [25], canine [26], and human [27] primary muscle satellite cells lost their stemness during the in vitro culture process. In this study, we also discovered that pig primary muscle satellite cells (pMuSCs) gradually lose their stemness during long-term in vitro passaging. By transducing lentiviral vectors expressing SV40 T-antigen, we were able to maintain the stemness of pMuSCs. However, compared to pMuSCs, SV40 T-pMuSCs exhibited a noticeable decrease in their differentiation capacity, and interference with the SV40 T-antigen gene could restore the normal differentiation ability.

Obtaining a sufficient number of functional pMuSCs requires significant labor and time, thus limiting studies involving pMuSCs. To understand the mechanism underlying the loss of stemness in satellite cells, we conducted a transcriptomic analysis of satellite cells at different passages. The transcriptional patterns were similar between primary satellite cells at P1 and P2 generations. However, a noticeable change in the transcriptional pattern was observed when comparing P4 with P1, and P4 with P2 generations (Figure 2A–C). Moreover, there was a significant reduction of the differentiation capacity of P10 generation pMuSCs compare with other early generations (Figure 1C). These results indicated that one should use lower passaging primary satellite cells for studying the pig’s muscle development and differentiation. 

Based on the transcriptomic results, we found that, in low-passage satellite cells, were mainly enriched in pathways such as MAPK signaling, PI3K-Akt signaling, and FoxO signaling. High-passage satellite cells showed a decline in abilities related to DNA replication, base excision repair, and mismatch repair. The cellular response to external stimuli requires the integration and activation of intracellular signaling pathways. These pathways largely determine post-translational modifications of proteins. Among them, phosphorylation being particularly important. One crucial pathway in intracellular signaling is the activation of mitogen-activated protein kinases (MAPKs), which participate in most cellular signaling pathways [28]. The forkhead box transcription factor (FoxO) family acts downstream of the PI3K/Akt pathway and regulates various physiological processes, including cell proliferation, survival, and metabolism [29]. It has been discovered that FoxO3 regulates the Notch signaling pathway, which is an important regulator of quiescence in adult satellite cells. The loss of FoxO3 results in decreased Notch signaling levels in satellite cells, indicating the significance of the FoxO3–Notch axis in promoting quiescence during self-renewal processes in satellite cells. Moreover, p38 is a subgroup of the MAPK family, and the p38 pathway is a major regulator of satellite cell fate determination, exerting positive effects at each stage of myogenesis [30]. The p38α/β MAPK signaling pathway orchestrates the asymmetric division of MuSCs. In this process, one daughter cell activates the p38α/β MAPK cascade, which in turn induces the expression of *MYOD* and leads to the generation of proliferative myoblasts. Conversely, in the other daughter cell, the p38α/β MAPK pathway remains inactive, and *MYOD* expression is not triggered, resulting in the production of quiescent MuSCs that help to preserve the stem cell reservoir [31]. The PI3K/AKT signaling pathway mediates the response of satellite cells to growth factors such as IGF1, which promotes key events in the regeneration process, including proliferation, muscle gene expression, myoblast fusion, survival, and post-mitotic growth of myotubes [32,33]. In contrast, high-generation satellite cells exhibit a decline in abilities such as DNA replication, base excision repair, and mismatch repair. DNA repair systems have evolved in eukaryotes to overcome DNA damage and include processes such as homologous recombination, non-homologous end joining, base excision repair, and nucleotide excision repair [34]. Studies have shown that factors involved in DNA repair also regulate cellular metabolism in response to DNA damage, thereby avoiding further genomic instability [35,36]. Ineffective DNA repair processes generate DNA fragments that enhance aging, which can be accelerated in cells lacking the homologous recombination repair system [37]. In summary, skeletal muscle satellite cells exhibit an elevated expression of MAPK signaling, PI3K-Akt signaling, and FoxO signaling at low passage numbers, which promote cell proliferation ability. However, as the cells undergo successive passages, there is a downregulation of DNA replication, base excision repair, and mismatch repair pathways. These findings suggest that prolonged in vitro cell culture weakens DNA repair capabilities, which may contribute to the decreased proliferative and differentiation potential of satellite cells. 

To overcome the decline in stemness of muscle satellite cells during long-term in vitro culture, we transduced pMuSCs with a lentiviral vector-expressing SV40 T-antigen. The results showed that the proliferation capacity of SV40 T-pMuSCs was maintained after SV40 T-antigen expression (Figure 3B). The binding of the SV40 T-antigen to the Rb protein inhibits pathways involved in cell cycle entry and growth arrest controlled by the Rb-E2F complex. E2F is a transcription factor that, together with Rb, regulates the transcription of E2F target genes, which encoded proteins required for DNA replication, nucleotide metabolism, DNA repair, and cell cycle progression. Disruption of the inhibitory effect of the SV40 T-antigen on the Rb-E2F complex leads to the transcription of E2F-dependent genes and entry into the S-phase, enabling continuous cell proliferation [18]. The SV40 T-antigen interacts with the tumor-suppressor protein p53, a transcriptional activator that mediates apoptosis under adverse conditions, thereby inhibiting cell death [38].

In addition, we found that the proportion of PAX7+ cells was initially low in SV40 T-pMuSCs at P1, and it gradually approached 60% with subsequent cell passages, similar to pMuSCs at P1 (Figure 3A). This may be due to the immediate suboptimal state of the cells upon SV40 T-antigen lentivirus infection, as the cells underwent 7 days of selection with hygromycin B, which could affect their normal gene expression patterns. Note that, when we performed the PAX7 immunostaining using P1-generation SV40 T-pMuSCs, we found that, the cell density was low, which could lead to a lower percentage of PAX7+ cells. This is consistent with the findings of Zheng et al., who demonstrated a significant increase in *PAX7* mRNA expression in satellite cells under high-density culture conditions [39]. As the cells continued to sub-culture and the cell density increased, their cellular state recovered, and the PAX7+ rate was also restored Figure 3A). Immunofluorescence staining of MyHC in differentiated SV40 T-pMuSCs at different generations revealed that the cells maintained their differentiation capacity (Figure 3D). However, the number of MyHC+ cells and the myotube fusion index were lower than that of pMuSCs at P1, suggesting that the SV40 T-antigen may affect the differentiation potential of pMuSCs. This result is consistent with the findings of Klundert et al., who discovered that the transformation of primary hamster myoblasts with the SV40 T-antigen led to the inhibition of terminal differentiation. This inhibition was associated with the transcriptional blockade of *MYOD* and *MYOG* [40]. 

To investigate the potential effects of SV40 T-antigen on the differentiation capacity of pMuSCs, we interfered with the expression of *SV40 T-antigen* and assessed the differentiation capacity of SV40 T-pMuSCs. Compared to the negative control (NC), the percentage of MyHC+ cells and fusion index were significantly increased after knockdown of SV40 T-antigen gene, although the fusion index remained slightly lower than that of pMuSCs. Activated satellite cells require cell cycle withdrawal and undergo asymmetric division, with one daughter cell undergoing differentiation [41]. Due to the continuous promotion of satellite cell proliferation by the SV40 T-antigen, it becomes difficult for cells to exit the cell cycle, thereby affecting their differentiation potential. Telomerase and CDK4 are two biologically significant molecules closely associated with cellular immortalization. Telomerase is an enzyme that maintains the length of telomeres during cell division. CDK4 (cyclin-dependent kinase 4) is a cell-cycle protein kinase that drives the progression of the cell cycle, allowing cells to bypass senescence and apoptosis, thereby achieving immortality [42]. Andrew et al. developed genetically immortalized bovine satellite cells (iBSCs) via the constitutive expression of bovine telomerase reverse transcriptase (TERT) and cyclin-dependent kinase 4 (CDK4), and maintain their capacity for myogenic differentiation [43]. This may represent an effective method for immortalizing pMuSCs without damaging their capacity for differentiation.

## 5. Conclusions

In summary, our study successfully isolated satellite cells from the longissimus dorsi muscle of 1-day-old Western commercial pigs and established SV40 T-pMuSCs by expressing the SV40 T-antigen gene. These cells have been sustained over 40 passages in vitro. Compared to primary P1-generation of pMuSCs, the differentiation potential of these cells was relatively weaker. Interference with SV40 T-antigen significantly enhances the differentiation capacity of the cells. This suggests that a conditional expression of SV40 T-antigen, such as the Tet-On system, may be is a better strategy to generate immortalized pMuSCs without jeopardizing its differentiation capacity. Nonetheless, the immortalized pMuSCs provide a valuable tool for investigating the role and mechanisms of satellite cells in skeletal muscle regeneration in pigs.

## Figures and Tables

**Figure 1 cells-13-00703-f001:**
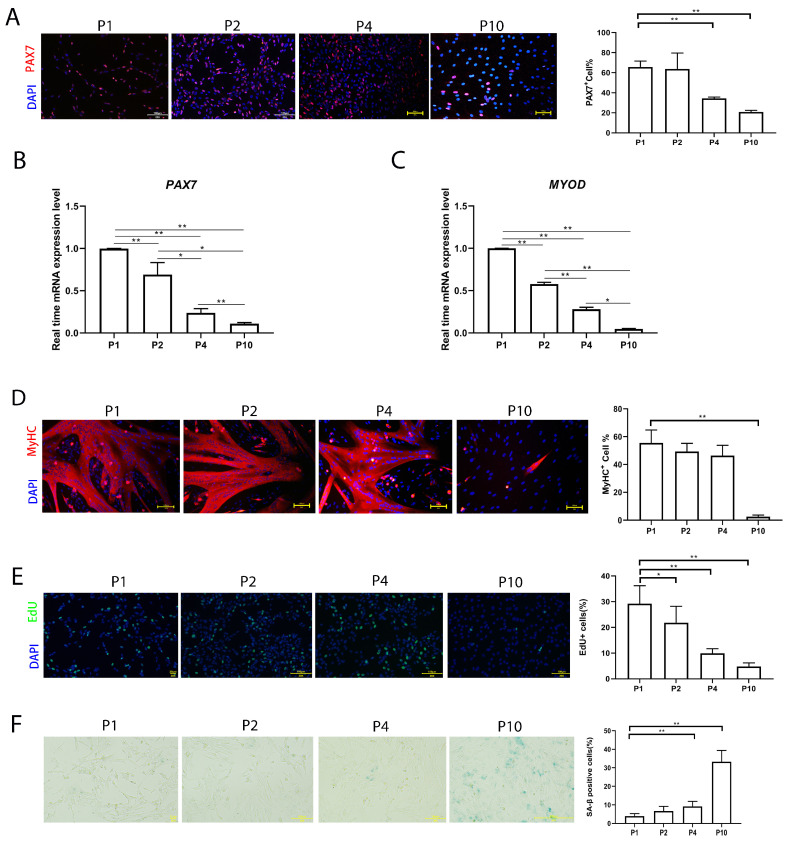
Evaluation of the function of pMuSCs during long-term in vitro culture. (**A**) Immunofluorescent staining of P1, P2, P4, and P10 pMuSCs using DAPI (blue) and PAX7 (red) antibody. Scale bar, 100 µm. (**B**) Relative mRNA expression of *PAX7* genes during in vitro sub-culturing. The expression level was normalized to *GAPDH* gene. (**C**) Relative mRNA expression of MYOD genes during in vitro sub-culturing. The expression level was normalized to GAPDH gene. (**D**) Immunofluorescent staining of differentiated myotubes from P1, P2, P4, and P10 pMuSCs using DAPI (blue) and anti-MyHC (red) antibody. Scale bar, 100 µm. (**E**) Staining of P1, P2, P4, and P10 pMuSCs using DAPI (blue) and EdU (green). Scale bar, 100 µm. (**F**) Representative images and quantitative results of SA-β-gal staining in P1, P2, P4, and P10 pMuSCs. Scale bar, 100 µm. One-way ANOVA with Tukey’s multiple comparison test was performed among different groups. All data were presented as mean ± SEM; * *p* < 0.05, ** *p* < 0.01.

**Figure 2 cells-13-00703-f002:**
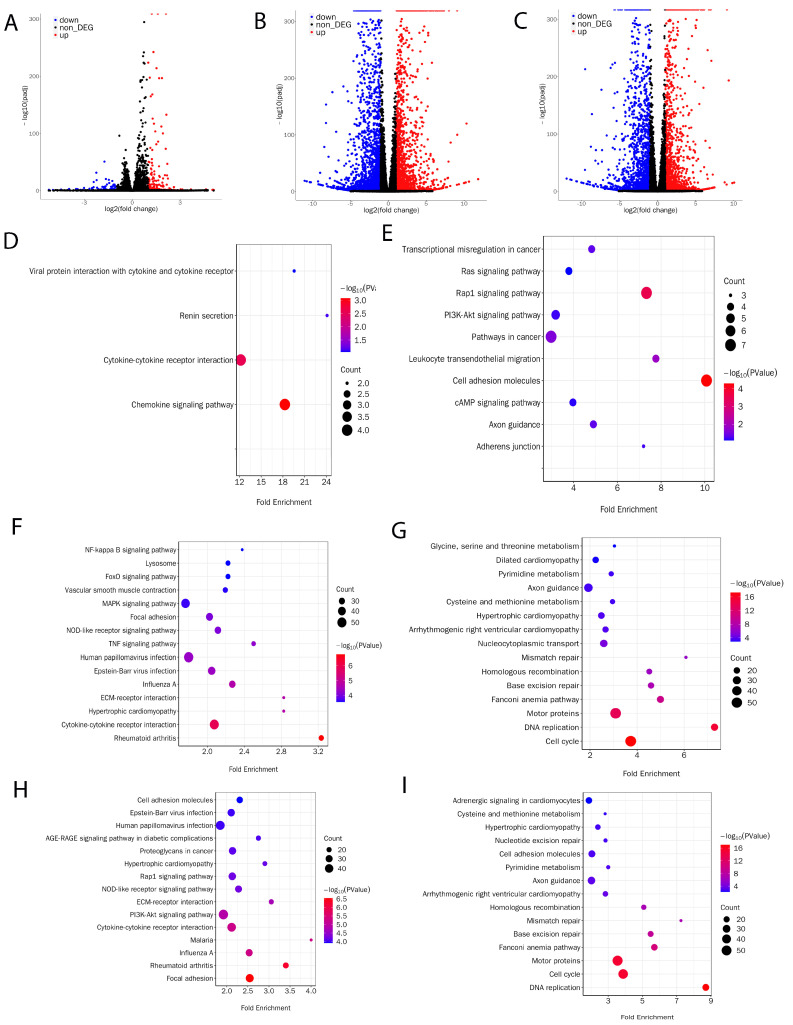
Evaluation of transcriptional profile of different passages of pMuSCs. Volcano plots showed the differentially expressed genes in pMuSCs between (**A**) P2 and P1 generations, (**B**) P4 and P1 generations, and (**C**) P4 and P2 generations. Bubble plots depict KEGG pathway enrichment analysis of significantly upregulated (**D**,**F**,**H**) or downregulated (**E**,**G**,**I)** genes in pMuSCs between P2 and P1 generations, P4 and P1 generations, and P4 and P2 generations, respectively. Padj < 0.001 and log2 (fold change) ≥ 1.

**Figure 3 cells-13-00703-f003:**
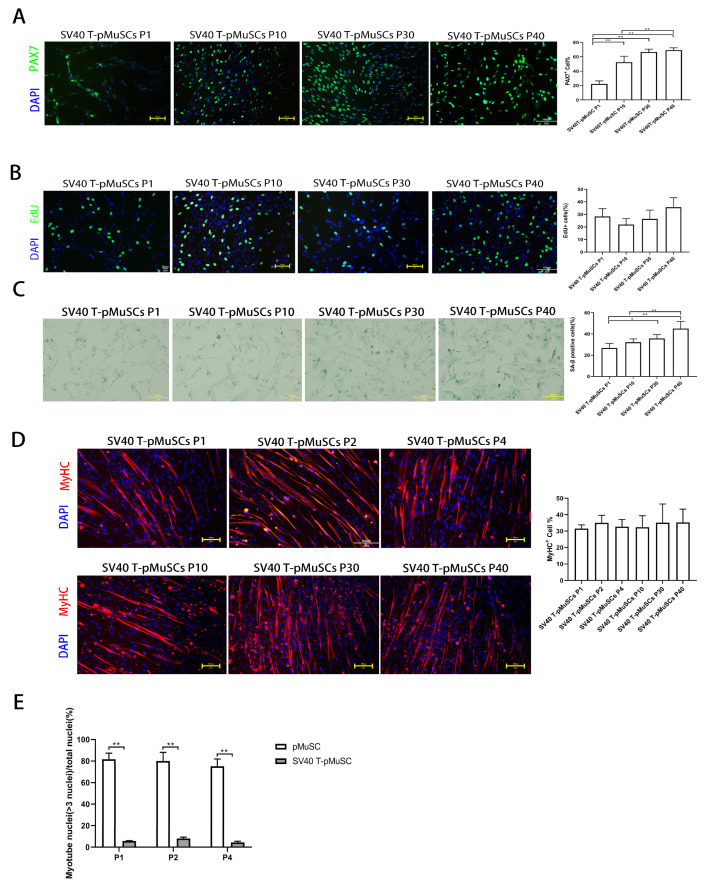
Establishment and characterization of SV40 T-pMuSCs. (**A**) Immunostaining of SV40 T-pMuSCs at P1, P10, P30, and P40 generations using DAPI (blue) and anti-PAX7 (green) antibody. Scale bar, 100 µm. (**B**) Immunostaining of SV40 T-pMuSCs at P1, P10, P30, and P40 generations using DAPI (blue) and EdU (green). Scale bar, 100 µm. (**C**) Representative images of SA-β-gal staining in SV40 T-pMuSCs at P1, P10, P30, and P40 generations. Scale bar, 100 µm. (**D**) Immunostaining of differentiated myotubes from SV40 T-pMuSCs at P1, P10, P30, and P40 generations using DAPI (blue) and anti-MyHC (red) antibody. Scale bar, 100 µm. (**E**) The myotube fusion index in primary and SV40 T-pMuSCs at P1, P2, and P4 generations. One-way ANOVA with Tukey’s multiple comparison test was performed among groups. All data were presented as mean ± SEM. * *p* < 0.05, ** *p* < 0.01.

**Figure 4 cells-13-00703-f004:**
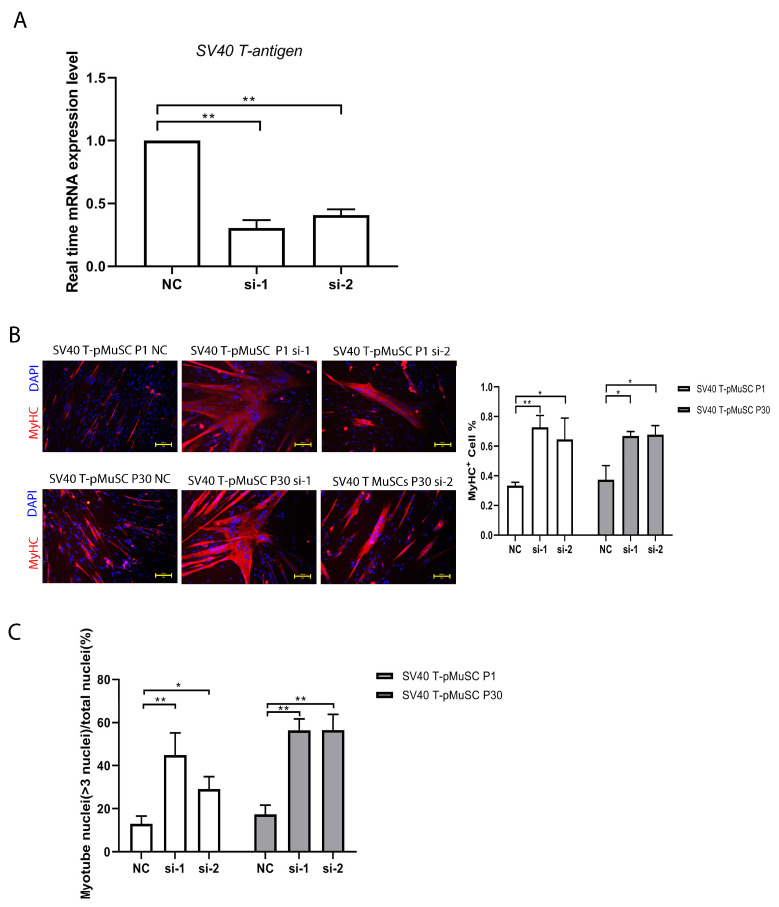
Interfering SV40 T-antigen restored the differentiation capacity of SV40 T-pMuSCs. (**A**) qPCR showed the siRNA knockdown efficiency of SV40 T-antigen. *GAPDH* transcript as a standard control. (**B**) The effect of knockdown of SV40 T-antigen on the differentiation capacity of P1 (**upper panel**) and P30 (**bottom panel**) generations of SV40 T-pMuSCs. DAPI (blue) and MyHC (red). Scale bar, 100 µm. (**C**) Quantification of the myotubes fusion index in P1 (left panel) and P30 (right panel) generation of SV40 T-pMuSCs in SV40 T-antigen gene knockdown conditions (si-1 and si-2) and control (NC). One-way ANOVA with Tukey’s multiple comparison test was performed among groups. All data were presented as mean ± SEM; * *p* < 0.05, ** *p* < 0.01.

## Data Availability

The data presented in this study are available upon request from the corresponding authors.

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
