# Peer review of "Establishment and Characterization of SV40 T-Antigen Immortalized Porcine Muscle Satellite Cell"

_cells, 2024, doi:10.3390/cells13080703_

Round 1
Reviewer 1 Report
Comments and Suggestions for Authors
Dear Authors,
Overall, I believe this study is well done and clearly written. The major focus is to demonstrate limitations in research using primary pig muscle satellite cells and how an immortalized line could possibly help overcome these limitations.
The authors provide valuable background information and describe the important role of pMuSC in ongoing research. They then proceed to demonstrate limitations in using primary cells, highlighting the rapid loss of proliferation and differentiation in these cells along with advancing cell senescence. They use immunohistochemistry and sequencing data to draw these conclusions.
This is followed by experiments to introduce SV40-T into the cell lineage to create an immortalized pMuSC line. They illustrate prolonged cell performance out to passage 40. They are also transparent with limitations in the cell line, highlighting limited Pax7 expression in early passages, lowered differentiation and fusion index data and increasing senescence with advancing passages. This is done using immunohistochemistry.
Lastly they show that differentiation can be improved by blocking SV40 expression at early and late passages using siRNA and observed with immunohistochemistry.
While I believe this study was well done. I have a few concerns.
Concern 1: The authors do an excellent job at illustrating limitations when using primary cells. In fact, they suggest that primary cell cultures should likely not go beyond passage 4 due to functional and gene expression differences that occur between passage 1/2 and 4. Based on this statement I am surprised that gene expression analysis was not used with subsequent experiments, especially after demonstrating limitation in cell functionality using immunohistochemistry (early Pax7 limitations, increasing cell senescence, and reduced differentiation). While they did partially address this using the siRNA blocking of SV40 to show a restoration of differentiation, would still have concerns about using these cells in an attempt to get the same outcomes as in early passage primary cells. Perhaps it is beyond the scope of this study, which is mostly focused on creating the immortalized cell line, but I do believe more examination of the cells would give greater confidence in using them for studies going forward.
Minor concern 2: Figure 4a has 3 significance bars, but only 3 groups.
Author Response
Thank you very much for taking the time to review this manuscript. Please find the detailed responses below and the corresponding revisions/corrections highlighted/in track changes in the re-submitted files.
|
Comments 1: The authors do an excellent job at illustrating limitations when using primary cells. In fact, they suggest that primary cell cultures should likely not go beyond passage 4 due to functional and gene expression differences that occur between passage 1/2 and 4. Based on this statement I am surprised that gene expression analysis was not used with subsequent experiments, especially after demonstrating limitation in cell functionality using immunohistochemistry (early Pax7 limitations, increasing cell senescence, and reduced differentiation). While they did partially address this using the siRNA blocking of SV40 to show a restoration of differentiation, would still have concerns about using these cells in an attempt to get the same outcomes as in early passage primary cells. Perhaps it is beyond the scope of this study, which is mostly focused on creating the immortalized cell line, but I do believe more examination of the cells would give greater confidence in using them for studies going forward.
|
|
Response 1: Thanks for your kind comments, your suggestion is valuable regarding to perform RNA-seq for SV40 T-antigen immortalized pig MuSCs and compare to the RNA-seq data for primary pig MuSCs. Thank you so much for pointing out this further direction for our study, we will do RNA-seq to evaluate its transcription pattern although these immortalized cell s showed some defect on differentiation. |
|
Comments 2: Figure 4a has 3 significance bars, but only 3 groups. |
|
Response 2: Thank you for pointing out this error, we have corrected it. |

Reviewer 2 Report
Comments and Suggestions for Authors
The authors describe the development of an immortalized porcine muscle satellite cell culture. The study is well performed and mainly well described, except some missing methodological details. Some minor revisions are recommended, as listed below, before the study can be published.
The title should end with “culture” or “model”
L78-79 needs a reference
L89 and 91 change “dorsal” to “dorsi”
L89 Clarify whether only one pig was used or if more than one, how many and if the cells were pooled for subsequent experiments.
L113 add information on number of replicates and time points of cultivation, when cells were used for immunohistochemical staining.
L123 add information on image capturing and analysis (camera, software, parameters).
L127 and 128 “IgG (H+L) (…” can be deleted
L131ff add information about samples and time points for RT-qPCR, “RT” is usually the abbreviation for “reverse transcription” not “real-time”
L141 add information on normalization and calculation of expression differences.
L143 better “Transfection of…” than “Transduce…”
L146 delete “transfection”
L156-157 How were the cells selected/sorted?
L161 Which samples, how many replicates were analyzed? How long were they cultured?
L163 correct the units
L168 details of analysis (software used, data obtained).
L180ff add information on samples used for sequencing.
L194-196 clarify which cells were harvested for RNA extraction and which were further cultivated for differentiation.
L211, 231, 267 and 279 I recommend changing “Evaluate the…” to “Evaluation of…”
L212 this is not a complete sentence
L267-277 abbreviations should be explained, gene names of swine should be written in capital letters.
Figure 1D correct the y-axis name
L320 the determination of the fusion index was not described in Material and Methods. Add respective information.
Figure 3B correct the y-axis name
Figure 4A add the gene name in the graph
L365 add some discussion whether this was also observed for primary MuSCs of other species.
L395-396 “transition from satellite cell” to what?
L417-423 needs a references
L441 add the respective reference number
L459 delete “can”
For all substances and devices, company, city, and country should be given. Names of porcine genes should be written with capital letters.
Supplementary Figure 1: correct “Lentiviral particle intection”
References in the reference list do not have a consistent format.
Comments on the Quality of English LanguageEnglish is mainly fine, only some minor changes are necessary.
Author Response
Thank you very much for your valuable comments on our manuscript. We greatly appreciate the opportunity to further revise our manuscript. We have checked the manuscript and revised it according to reviewers comments. We submit here the revised version as well as the point-by-point response for your review. All modifications have been tracked under the revisions mode. Line numbers in “Response to Reviewers” file correspond to the line numbers in revised Manuscript.
|
Comments 1: The title should end with “culture” or “model” |
|
Response 1: Thanks for your suggestion, instead of adding “culture” or “model” in the end of the title, we added “line”, we hope you could agree with this change, thanks (L3).
|
|
Comments 2: L78-79 needs a reference. |
|
Response 2: Thanks for your comment. We have supplemented the references, please see in L79.
|
|
Comments 3: L89 and 91 change “dorsal” to “dorsi” |
|
Response 3: Thanks for your correction, we have changed these words in main text, see in L90-92.
Comments 4: Clarify whether only one pig was used or if more than one, how many and if the cells were pooled for subsequent experiments. |
|
Response 4: Thanks for your suggestion, we used one 1-day-old Western commercial pigs. We had added this information in L91.
Comments 5: add information on number of replicates and time points of cultivation, when cells were used for immunohistochemical staining. Response 5: Thanks for your suggestion, we have added these informations in L119-120.
Comments 6: add information on image capturing and analysis (camera, software, parameters). Response 6: Thanks for your suggestion, we had added these information (L131).
Comments 7: L127 and 128 “IgG (H+L) (…” can be deleted Response 7: Thanks for your suggestion, we had deleted these (L136-138).
Comments 8: L131ff add information about samples and time points for RT-qPCR, “RT” is usually the abbreviation for “reverse transcription” not “real-time” Response 8: Thanks for your suggestion, we have added these informations on L140, and changed “RT-qPCR” to “real-time quantitative PCR”.
Comments 9: L141 add information on normalization and calculation of expression differences. Response 9: Thanks for your suggestion, we have provided additional information regarding the normalization and calculation of expression differences (L158-161).
Comments 10: L143 better “Transfection of…” than “Transduce…” Response 10: Thanks for your suggestion, since we used lentivirus plasmid, we changed it to “Transduction of”, we hope you could agree with this change, thanks (L163).
Comments 11: L146 delete “transfection” Response 11: Thanks for your suggestion, we have deleted the word, sorry for the mistake (L166).
Comments 12: L156-157 How were the cells selected/sorted? Response 12: Thanks for your suggestion, we selected the positive cell using 200μg/ml hygromycin B (antibiotic) for 7 days (L174-175).
Comments 13: L161 Which samples, how many replicates were analyzed? How long were they cultured? Response 13: Thanks for your suggestion, we have provided these detail information regarding the EdU assay. (L183-184, L186-187).
Comments 14: L163 correct the units. Response 14: Thanks for your suggestion, upon reviewing the kit instructions, we confirmed that the final concentration of 1× EdU is 10 µM.
Comments 15: L168 details of analysis (software used, data obtained). Response 15: T hanks for your suggestion, we have supplemented the details on image acquisition. (L192-194)
Comments 16: L180ff add information on samples used for sequencing. Response 16: Thanks for your suggestion, we have supplemented the information regarding the samples used for sequencing (L207-208).
Comments 17: L194-196 clarify which cells were harvested for RNA extraction and which were further cultivated for differentiation. Response 17: Thanks for your suggestion, we have added a description of the cell for RNA extraction and the subsequent differentiation of the cells (L219-221, L226-228).
Comments 18: L211, 231, 267 and 279 I recommend changing “Evaluate the…” to “Evaluation of…” Response 18: We have changed to “Evaluation of”, see (L242-243, L299), thanks.
Comments 19: L212 this is not a complete sentence. Response 19: Thanks for your suggestion, we have refined the grammar of this sentence (L243-245).
Comments 20: L267-277 abbreviations should be explained, gene names of swine should be written in capital letters. Response 20: Thanks for your suggestion, the full name of the abbreviation pMuSCs is introduced in Line 16, MyHC is detailed in Line 46, and the full term for EdU is introduced in Line 184. The term SA-β-gal is explained in Lines 198-1909 I have capitalized the MYOD and PAX7 gene in full text and the images.
Comments 21: Figure 1D correct the y-axis name. Response 21: Thanks for your suggestion, we have corrected they-axis name EdU+ cells (%) in Figure 1D.
Comments 22: L320 the determination of the fusion index was not described in Material and Methods. Add respective information. Response 22: Thanks for your suggestion, we have supplemented the description of the myotube fusion index determination method in the Materials and Methods section (L113-116).
Comments 23: Figure 3B correct the y-axis name. Response 23: Thanks for your suggestion, we have corrected they-axis name to “EdU+ cells (%)” in Figure 3B.
Comments 24: Figure 4A add the gene name in the graph. Response 24: Thanks for your suggestion, we have added the gene name “SV40 T-antigen” in Figure 4A.
Comments 25: L365 add some discussion whether this was also observed for primary MuSCs of other species. Response 25: Thanks for your suggestion, we have added discussions regarding muscle satellite cells from other species, along with relevant references, have been included (L396-398).
Comments 26: L395-396 “transition from satellite cell” to what? Response 26: Thanks for your suggestion, we have included the corresponding discussions and references to supplement the information provided (L430-435).
Comments 27: L417-423 needs a reference. Response 27: Thanks for your suggestion, we have added the reference (L462).
Comments 28: L441 add the respective reference number. Response 28: Thanks for your suggestion, we have added the reference (L482).
Comments 29: L459 delete “can”. Response 29: Thanks for your suggestion, we have deleted the word (L498).
|
Comments 30: For all substances and devices, company, city, and country should be given. Names of porcine genes should be written with capital letters.
Response 30: The text has been revised to include the capitalization of porcine gene names throughout. Additionally, the countries of origin for some reagents and equipment have been provided, however, for certain pieces of equipment, we did not specify both the country and city because the official websites do not have these informations. Hope you could understand for missing some of these informations. Thanks.
Comments 31: Supplementary Figure 1: correct “Lentiviral particle intection”.
Response 31: Thanks for your suggestion, we have corrected as” Lentiviral particle infection”, we have made an additional correction to a spelling error in a word.
Comments 32: References in the reference list do not have a consistent format.
Response 32: Thanks for your suggestion, we have standardized the format of the references.
